# $(q, \sigma, \tau)$-Differential Graded Algebras

**Viktor Abramov [1,*]**, **Olga Liivapuu [2]** and **Abdenacer Makhlouf [3]**

1   Institute of Mathematics and Statistics, University of Tartu, J. Liivi 2, 50409 Tartu, Estonia
2   Institute of Technology, University of Life Sciences, Kreutzwaldi 1, 51006 Tartu, Estonia;
    olga.liivapuu@emu.ee
3   IRIMAS—Département de Mathématiques, Université de Haute Alsace, F-68093 Mulhouse, France;
    abdenacer.makhlouf@uha.fr
*   Correspondence: viktor.abramov@ut.ee; Tel.: +372-737-5862

**Abstract:** We propose the notion of $(q, \sigma, \tau)$-differential graded algebra, which generalizes the notions of $(\sigma, \tau)$-differential graded algebra and $q$-differential graded algebra. We construct two examples of $(q, \sigma, \tau)$-differential graded algebra, where the first one is constructed by means of the generalized Clifford algebra with two generators (reduced quantum plane), where we use a $(\sigma, \tau)$-twisted graded $q$-commutator. In order to construct the second example, we introduce the notion of $(\sigma, \tau)$-pre-cosimplicial algebra.

**Keywords:** $q$-differential graded algebra; $(\sigma, \tau)$-differential graded algebra; generalized Clifford algebra; pre-cosimplicial complex

## 1. Introduction

Skew-derivations or $\sigma$-derivations are generalized derivations obtained by twisting the Leibniz rule by means of an algebra map. They have been considered by physicists to study quantum groups and to obtain $q$-deformations of algebra of vector fields like Virasoro algebra and also Heisenberg algebras (oscillator algebras); see [1–5]. The main example is given by the Jackson derivative and leads for example to $q$-deformation of sl$_2$, Witt algebra, and Virasoro algebra. A natural generalization consists of $(\sigma, \tau)$-derivations involving two twist maps [6]. It turns out that when using $\sigma$-derivations, the commutator bracket no longer satisfies the Jacobi condition. This was a starting point of studying hom-type algebras, where the usual identities are twisted by homomorphisms [7,8]. Later development in the field of research of hom-Lie algebras led to the introduction of the notion of $(\sigma, \tau)$-differential graded algebra [9], which generalizes the notion of differential graded algebra.

It is well known that the concept of differential graded algebra is based on the equation $d^2 = 0$, where $d$ is a differential of differential graded algebra. In order to generalize the concept of differential graded algebra, one can consider instead of $d^2 = 0$ a more general equation $d^N = 0$, where $N$ is an integer greater than or equal to two. This generalization was proposed and studied in [10]. We would like to mention that equation $d^N = 0$ leads to generalized cohomologies, which can be applied in the quantum Wess–Zumino–Novikov–Witten (WZNW) model for the realization of the space of physical states [11]. Later, an algebraic structure, based on the equation $d^N = 0$, was developed in [12], where the authors proposed the notion of $q$-differential graded algebra, where $q$ is a primitive $N^{\text{th}}$ root of unity. It is worth mentioning that particularly in the case of $q = -1$ (primitive square root of unity, $N = 2$), the definition of $q$-differential graded algebra gives the definition of differential graded algebra. Later, it was shown that $q$-differential graded algebras can be applied in the field theories [13] and in noncommutative geometry to develop a generalization of the notion of connection [14].

In this paper, we propose and study the notion of $(q, \sigma, \tau)$-differential algebra, where $q$ is the primitive $N^{\text{th}}$ root of unity and $\sigma, \tau$ two degree zero endomorphisms of a graded algebra. This notion generalizes the notion of $(\sigma, \tau)$-differential graded algebra, introduced in [9], as well as the notion of $q$-differential graded algebra, which was studied in [15–17]. We construct several examples of $(q, \sigma, \tau)$-differential graded algebras by means of the generalized Clifford algebra with two generators (also called a reduced quantum plane) and by means of pre-cosimplicial algebra.

## 2. First Order $(\sigma, \tau)$-Differential Calculus with Right Partial Derivatives

Let $\mathfrak{A}$ be an associative unital algebra over $\mathbb{K}$, where $\mathbb{K}$ is either the field of real $\mathbb{R}$ or complex numbers $\mathbb{C}$. Let $\sigma, \tau$ be two algebra endomorphisms of $\mathfrak{A}$.

**Definition 1.** *A first order $(\sigma, \tau)$-differential calculus over an algebra $\mathfrak{A}$ is the triple $(\mathfrak{A}, d, \mathfrak{M})$, where $\mathfrak{M}$ is an $(\mathfrak{A}, \mathfrak{A})$-bimodule and $d : \mathfrak{A} \to \mathfrak{M}$ is a linear mapping, which satisfies the $(\sigma, \tau)$-Leibniz rule:*

$$d(uv) = d(u)\,\tau(v) + \sigma(u)\,d(v). \tag{1}$$

In particular, if $\sigma, \tau$ are the identity transformations of algebra $\mathfrak{A}$, i.e., $\sigma = \tau = \text{id}_{\mathfrak{A}}$, then the notion of first order $(\sigma, \tau)$-differential calculus amounts to the notion of first order differential calculus over an associative unital algebra.

Now, we assume that $(\mathfrak{A}, d, \mathfrak{M})$ is a first order $(\sigma, \tau)$-differential calculus, where $\mathfrak{M}$ is a free finite right $\mathfrak{A}$-module of rank $r$ with a basis $e^1, e^2, \ldots, e^r$, i.e., any element $u$ of $\mathfrak{M}$ can be uniquely written as:

$$u = \sum_{i=1}^{r} e^i\, u_i, \quad u_i \in \mathfrak{A}.$$

Then, a structure of the $(\mathfrak{A}, \mathfrak{A})$-bimodule of $\mathfrak{M}$ is uniquely determined by the commutation relations:

$$v\,e^i = \sum_{j=1}^{r} e^j\, R_j^i(v), \tag{2}$$

where the linear mappings $R_j^i : \mathfrak{A} \to \mathfrak{A}$ satisfy:

$$R_j^i(uv) = R_j^k(u)\, R_k^i(v). \tag{3}$$

It is useful to compose the $r^{\text{th}}$ order square matrix $R = (R_j^i)$ such that a linear mapping $R_j^i$ is its entry at the intersection of the $i^{\text{th}}$ column and $j^{\text{th}}$ row. Then, (3) can be written in matrix form as follows $R(uv) = R(u)\,R(v)$. Hence, the linear mapping $R : \mathfrak{A} \to M_r(\mathfrak{A})$, where $M_r(\mathfrak{A})$ is the algebra of $r^{\text{th}}$ order matrices over an algebra $\mathfrak{A}$, is the algebra homomorphism. Define the right partial derivatives $\partial_i : \mathfrak{A} \to \mathfrak{A}$ (in a basis $e^i$) by the formula:

$$du = \sum_i e^i\, \partial_i(u). \tag{4}$$

**Proposition 1.** *If $(\mathfrak{A}, d, \mathfrak{M})$ is a first order $(\sigma, \tau)$-differential calculus over an algebra $\mathfrak{A}$ and $\mathfrak{M}$ is a free finite right $\mathfrak{A}$-module with a basis $e^1, e^2, \ldots, e^n$, whose $(\mathfrak{A}, \mathfrak{A})$-bimodule structure is determined by the commutation relations (2), then the right partial derivatives, defined in (4), satisfy:*

$$\partial_i(uv) = \partial_i(u)\,\tau(v) + \sum_j R_i^j(\sigma(u))\,\partial_j(v). \tag{5}$$

**Proof.** According to the definition of right partial derivatives, we can write:

$$d(uv) = \sum_i e^i \partial_i(uv).$$

On the other hand, making use of the $(\sigma, \tau)$-Leibniz rule, we get:

$$
\begin{aligned}
d(uv) &= d(u)\,\tau(v) + \sigma(u)\,d(v) = \sum_i \left(e^i\,\partial_i(u)\right)\tau(v) + \sigma(u)\sum_j e^j\,\partial_j(v) \\
&= \sum_i \left(e^i\,\partial_i(u)\right)\tau(v) + \sum_{i,j} e^i\,R_i^j(\sigma(u))\,\partial_j(v) = \sum_i e^i\,(\partial_i(u))\,\tau(v) + R_i^j(\sigma(u))\,\partial_j(v)).
\end{aligned}
$$

□

A first order $(\sigma, \tau)$-differential calculus $(\mathfrak{A}, d, \mathfrak{M})$, where $\mathfrak{M}$ is a free right $\mathfrak{A}$-module of rank $r$, whose $(\mathfrak{A}, \mathfrak{A})$-bimodule structure is determined by the commutation rule (2) and the right derivatives are defined by (4), will be referred to as a first order $(\sigma, \tau)$-differential calculus with right partial derivatives. If $(\mathfrak{A}, d, \mathfrak{M})$ is a first order $(\sigma, \tau)$-differential calculus with right partial derivatives, an algebra $\mathfrak{A}$ is generated by variables $x^1, x^2, \ldots, x^r$, and $dx^1, dx^2, \ldots, dx^r$ are the basis for a free right $\mathfrak{A}$-module $\mathfrak{M}$, then this first order $(\sigma, \tau)$-differential calculus will be referred to as a coordinate first order $(\sigma, \tau)$-differential calculus with right partial derivatives.

### 3. $(q, \sigma, \tau)$-Differential Graded Algebra

Let $\mathcal{A} = \oplus_n \mathcal{A}^n$ be a graded associative unital algebra and $\sigma, \tau$ be degree zero endomorphisms of $\mathcal{A}$. The degree of a homogeneous element $u$ will be denoted by $|u|$. In what follows, $q$ will be a primitive $N^{\text{th}}$ root of unity, where $N \geq 2$. For instance, we can take $q = \exp(2\pi i / N)$. We give the following definition:

**Definition 2.** *$\mathcal{A}$ is said to be a $(q, \sigma, \tau)$-differential graded algebra if $\mathcal{A}$ is endowed with a degree one linear mapping $d : \mathcal{A}^n \to \mathcal{A}^{n+1}$ such that it satisfies the following conditions:*

(a)  *$d$ commutes with endomorphisms $\sigma, \tau$, i.e., $\sigma \circ d = d \circ \sigma, \tau \circ d = d \circ \tau$,*

(b)  *$d$ satisfies the $(q, \sigma, \tau)$-Leibniz rule:*

$$d(uv) = d(u)\,\tau(v) + q^{|u|}\,\sigma(u)\,d(v),$$

(c)  *$d^N(u) = 0$ for any element $u \in \mathcal{A}$.*

In particular, if we choose in the above definition $q = -1 (N = 2)$ and $\sigma = \tau = \mathrm{id}_{\mathcal{A}}$, then we get the definition of differential graded algebra. If we fix $\sigma = \tau = \mathrm{id}_{\mathcal{A}}$, but $N$ is an arbitrary integer greater than two, then the notion of $(q, \sigma, \tau)$-differential graded algebra gives the notion of $q$-differential graded algebra [12–18]. If we put $N = 2$ (then $q = -1$) and $\sigma, \tau$ are different from $\mathrm{id}_{\mathcal{A}}$, then the definition of $(q, \sigma, \tau)$-differential graded algebra gives the definition of $(\sigma, \tau)$-differential algebra [9]. An $N^{\text{th}}$ root of unity $q$ determines a graded structure of $(q, \sigma, \tau)$-differential graded algebra, and when it increases (then $q \to 1$), a graded structure of $(q, \sigma, \tau)$-differential graded algebra differs more and more from the classical $\mathbb{Z}_2$-graded structure of differential graded algebra, because there appear more and more subspaces of different degrees, which can be labeled by integers $0, 1, \ldots, N - 1$.

Let $\mathcal{A} = \oplus_n \mathcal{A}^n$ be a $(q, \sigma, \tau)$-differential graded algebra. Obviously, $\mathcal{A}^0 \subset \mathcal{A}$ is the subalgebra of $\mathcal{A}$. Next, it is easy to see that every subspace of homogeneous elements $\mathcal{A}^n$ can be considered as an $(\mathcal{A}^0, \mathcal{A}^0)$-bimodule, where the left (right) $\mathcal{A}^0$-module structure of $\mathcal{A}^n$ is determined by the left (right) multiplication by degree zero elements, i.e.,

$$(u, v) \in \mathcal{A}^0 \times \mathcal{A}^n \to u\,v \in \mathcal{A}^n, \quad (v, u) \in \mathcal{A}^n \times \mathcal{A}^0 \to v\,u \in \mathcal{A}^n.$$

Hence, the triple $(\mathcal{A}^0, d, \mathcal{A}^1)$ is the first order $(\sigma, \tau)$-differential calculus, because $\mathcal{A}^1$ is the $(\mathcal{A}^0, \mathcal{A}^0)$-bimodule and $d : \mathcal{A}^0 \to \mathcal{A}^1$ satisfies in this case the $(\sigma, \tau)$-Leibniz rule:

$$d(uv) = d(u)\,\tau(v) + \sigma(u)\,d(v), \quad u, v \in \mathcal{A}^0.$$

## 4. Construction of $(q, \sigma, \tau)$-Differential Graded Algebra by Means of the Graded $q$-Commutator

In this section, we show that given an associative unital graded algebra, one can construct a $(q, \sigma, \tau)$-differential graded algebra with the help of the graded $q$-commutator, where $q$ is a primitive $N^{\text{th}}$ root of unity.

Let $\mathcal{A} = \oplus_n \mathcal{A}^n$ be a graded algebra over $\mathbb{C}$, $\mathbf{1}$ be its unit element, and $\sigma, \tau$ be two degree zero endomorphisms of $\mathcal{A}$.

**Theorem 1.** *Let $\xi \in \mathcal{A}^1$ be an element of degree one. Define the degree one linear mapping $d : \mathcal{A}^n \to \mathcal{A}^{n+1}$ by the following formula:*

$$d(u) = \xi\,\tau(u) - q^{|u|}\sigma(u)\,\xi. \tag{6}$$

*If:*

(a)  *$q$ is a primitive $N^{\text{th}}$ root of unity,*
(b)  *$\xi^N = \lambda\,\mathbf{1}$, where $\lambda$ is a non-zero complex number,*
(c)  *$\sigma(\xi) = \tau(\xi) = \xi$, $\sigma \circ \tau = \tau \circ \sigma$, $\sigma^N = \tau^N$,*

*then a graded algebra $\mathcal{A}$ endowed with the degree one linear mapping $d$ is the $(q, \sigma, \tau)$-differential graded algebra.*

**Proof.** First we prove $\sigma \circ d = d \circ \sigma$. For any homogeneous $u \in \mathcal{A}$, we have:

$$
\begin{aligned}
\sigma \circ d(u) &= \sigma\big(\xi\,\tau(u) - q^{|u|}\sigma(u)\,\xi\big) = \sigma(\xi)\,\sigma \circ \tau(u) - q^{|u|}\,\sigma^2(u)\,\sigma(\xi) \\
&= \xi\,\tau \circ \sigma(u) - q^{|u|}\sigma^2(u)\,\xi = d \circ \sigma(u).
\end{aligned}
$$

Analogously, we can prove $\tau \circ d = d \circ \tau$. Starting with the right-hand side of the $(q, \sigma, \tau)$-Leibniz rule (Definition 2), we get:

$$
\begin{aligned}
d(u)\,\tau(v) + q^{|u|}\,\sigma(u)\,d(v) &= \big(\xi\,\tau(u) - q^{|u|}\sigma(u)\,\xi\big)\tau(v) + q^{|u|}\,\sigma(u)\big(\xi\,\tau(v) - q^{|v|}\,\sigma(v)\,\xi\big) \\
&= \xi\,\tau(uv) - q^{|u|}\sigma(u)\,\xi\,\tau(v) + q^{|u|}\sigma(u)\,\xi\,\tau(v) - q^{|u|+|v|}\sigma(uv)\,\xi = d(uv),
\end{aligned}
$$

and the $(q, \sigma, \tau)$-Leibniz rule is proven. For the $N^{\text{th}}$ power of $d$, we have the following power expansion (see [15]):

$$d^N(u) = \sum_{i=0}^{N} (-1)^i\,p_i \begin{bmatrix} N \\ i \end{bmatrix}_q \xi^{N-i}\,\tau^{N-i} \circ \sigma^i(u)\,\xi^i, \tag{7}$$

where $p_i = q^{i|u|+\mu(i)}$ and $\mu(i) = \frac{i(i-1)}{2}$. According to the assumption, $q$ is a primitive $N^{\text{th}}$ root of unity, which implies for the quantum Newton binomial coefficients:

$$\begin{bmatrix} N \\ i \end{bmatrix}_q = 0, \quad i = 1, 2, \ldots, N-1,$$

and the terms in the power expansion (7) labeled by $i = 1, 2, \ldots, N-1$ vanish. Thus, there are only two non-trivial terms in (7):

$$d^N(u) = \lambda\left(\tau^N(u) + (-1)^N\,q^{N(N-1)/2}\sigma^N(u)\right).$$

If $N$ is an odd positive integer, then:

$$d^N(u) = \lambda \left( \tau^N(u) - (q^N)^{\frac{N-1}{2}} \sigma^N(u) \right) = \lambda \left( \tau^N(u) - \sigma^N(u) \right) = 0.$$

If $N$ is an even positive integer, then:

$$d^N(u) = \lambda \left( \tau^N(u) + (q^{\frac{N}{2}})^{N-1} \right) = \lambda \left( \tau^N(u) + (-1)^{N-1} \sigma^N(u) \right) = \lambda \left( \tau^N(u) - \sigma^N(u) \right) = 0.$$

□

In order to construct a matrix example of $(q, \sigma, \tau)$-differential graded algebra, we apply Theorem 1 to the generalized Clifford algebra. We recall you that the generalized Clifford algebra is an associative unital algebra over $\mathbb{C}$ generated by variables $x^1, x^2, \ldots, x^n$, which are subjected to the relations:

$$x^i x^j = q \, x^j x^i \;\; (i < j), \;\; (x^i)^N = \mathbf{1}, \;\; i, j = 1, 2, \ldots, n, \tag{8}$$

where $q$ is a primitive $N^{\text{th}}$ root of unity and $\mathbf{1}$ is the unit element of the generalized Clifford algebra. The generalized Clifford algebra will be denoted by $\mathfrak{C}_n^N$, where $n, N$ are independent integers, $n$ is the number of generators, and $N \geq 2$ is an exponent at which the $N^{\text{th}}$ power of every generator equals the identity element $\mathbf{1}$.

We consider the generalized Clifford algebra $\mathfrak{C}_2^N$ with two generators $x = x^1, \xi = x^2$. Then, from the relations (8), it follows:

$$x \xi = q \, \xi x, \;\; x^N = \xi^N = \mathbf{1}. \tag{9}$$

The associative unital algebra generated by two variables $x, \xi$, which are subjected to the relations (9), is also called the algebra of functions on a reduced quantum plane. This algebra has the matrix representation by $N^{\text{th}}$ order complex matrices. Indeed, we can identify the generators $x, \xi$ with the Weyl pair, i.e., two $N^{\text{th}}$ order matrices:

$$x = \begin{pmatrix} 1 & 0 & 0 & \ldots & 0 \\ 0 & q^{-1} & 0 & \ldots & 0 \\ 0 & 0 & q^{-2} & \ldots & 0 \\ \ldots & \ldots & \ldots & \ldots & \ldots \\ 0 & 0 & 0 & \ldots & q^{-(N-1)} \end{pmatrix}, \;\; \xi = \begin{pmatrix} 0 & 1 & 0 & \ldots & 0 \\ 0 & 0 & 1 & \ldots & 0 \\ \ldots & \ldots & \ldots & \ldots & \ldots \\ 0 & 0 & 0 & \ldots & 1 \\ 1 & 0 & 0 & \ldots & 0 \end{pmatrix}. \tag{10}$$

Since the Weyl pair satisfies the relations (9), it provides the matrix representation for the generalized Clifford algebra with two generators. It is worth mentioning that the matrices (10) generate the whole algebra of $N^{\text{th}}$ order complex matrices $M_N(\mathbb{C})$, and they are widely used in quantum information processing theory [19].

In order to define a structure of graded algebra on $\mathfrak{C}_2^N$, we attribute degree zero to the unit element $\mathbf{1}$ and the generator $x$, degree one to the generator $\xi$, and extend this degree to any product of the generators $x, \xi$ by defining the degree of a product as the sum of degrees of its cofactors. Then, the whole algebra $\mathfrak{C}_2^N$ splits into the direct sum of subspaces of homogeneous elements, and a subspace of elements of degree $k$ will be denoted by $\mathfrak{C}_2^{N,k}$, where $k$ runs over the residue classes modulo $N$, i.e., $k = 0, 1, \ldots, N - 1$. Obviously, the subalgebra of elements of degree zero $\mathfrak{C}_2^{N,0}$ will be generated by the generator $x$, i.e., $\mathfrak{C}_2^{N,0}$ is the algebra of polynomials of $x$. We consider the generator $x$ as an analog of the coordinate function of one-dimensional space. Thus, the algebra of "functions" is the algebra of polynomials of $x$. In order to emphasize that we consider the elements of $\mathfrak{C}_2^{N,0}$ as analogs of functions, we will denote the elements of $\mathfrak{C}_2^{N,0}$ by $f(x), g(x), h(x)$.

Let $\sigma, \tau$ be two degree zero endomorphisms of the generalized Clifford algebra $\mathfrak{C}_2^N$ such that they commute, $\sigma(\xi) = \tau(\xi) = \xi$, $\sigma^N = \tau^N$, and $\tau(x) - q\,\sigma(x)$ is the invertible element of $\mathfrak{C}_2^{N,0}$. According to Theorem 1, we define the differential $d_\xi : \mathfrak{C}_2^{N,k} \to \mathfrak{C}_2^{N,k+1}$ by the formula:

$$d_\xi(u) = \xi\,\tau(u) - q^{|u|}\,\sigma(u)\,\xi, \tag{11}$$

where $u$ is a homogeneous element of the generalized Clifford algebra $\mathfrak{C}_2^N$ and $|u|$ is its degree. Since all the assumptions of Theorem 1 are fulfilled, the algebra of $N^{\text{th}}$ order complex matrices $M_N(\mathbb{C})$ endowed with the structure of $\mathbb{Z}_N$-graded algebra, which is based on $|x| = 0, |\xi| = 1$, and with the differential $d_\xi$ is the $(q, \sigma, \tau)$-differential graded algebra.

We conclude this section by considering the structure of the first order differential calculus of matrix $(q, \sigma, \tau)$-differential graded algebra $\mathfrak{C}_2^N$. It is easy to show that $(\mathfrak{C}_2^{N,0}, d_\xi, \mathfrak{C}_2^{N,1})$ is the coordinate first order $(\sigma, \tau)$-differential calculus with right derivative. Indeed, we have:

$$d_\xi(x) = \xi\,\tau(x) - \sigma(x)\xi = \xi\left(\tau(x) - q\,\sigma(x)\right).$$

Because we assume that $\tau(x) - q\,\sigma(x)$ is an invertible element of $\mathfrak{C}_2^{N,0}$, the differential $d_\xi(x)$ of coordinate function $x$ can serve as the basis for the right $\mathfrak{C}_2^{N,1}$-module. The commutation relation, which determines the $\mathfrak{C}_2^{N,0}$-bimodule structure of $\mathfrak{C}_2^{N,1}$, has the form:

$$f(x)\,dx = dx\,R(f(x)),$$

where $R : f(x) \in \mathfrak{C}_2^{N,0} \to f(qx) \in \mathfrak{C}_2^{N,0}$ is the automorphism of the algebra. Thus, according to (4), the differential $d_\xi$ induces the right derivative:

$$d_\xi f(x) = d_\xi x\,\frac{df(x)}{dx}, \tag{12}$$

where $f$ is a polynomial of $x$. According to Proposition 1, this derivative satisfies:

$$\frac{d}{dx}(f(x)g(x)) = \frac{df(x)}{dx}\tau(g(x)) + \sigma(f(qx))\,\frac{dg(x)}{dx}. \tag{13}$$

Since $\sigma, \tau$ are linear mappings, the left-hand side of (12) can be written as:

$$
\begin{aligned}
d_\xi f(x) &= \xi\,\tau(f(x)) - \sigma(f(x))\,\xi = \xi\left(f(\tau(x)) - f(\sigma(qx))\right) \\
&= \xi\left(\tau(x) - q\,\sigma(x)\right)\frac{f(\tau(x)) - f(\sigma(qx))}{\tau(x) - \sigma(qx)} = d_\xi x\,\frac{f(\tau(x)) - f(\sigma(qx))}{\tau(x) - \sigma(qx)},
\end{aligned}
$$

where:

$$\left(\tau(x) - \sigma(qx)\right)^{-1} = \frac{1}{\tau(x) - \sigma(qx)}$$

is the inverse of $\tau(x) - \sigma(qx)$. From this, it follows:

$$\frac{df(x)}{dx} = \frac{f(\tau(x)) - f(\sigma(qx))}{\tau(x) - \sigma(qx)}. \tag{14}$$

The derivative (14) is called the $(\sigma, \tau)$-twisted Jackson $q$-derivative [7]. Thus, the differential $d_\xi$, defined in (11), determines the coordinate first order $(\sigma, \tau)$-differential calculus with the $(\sigma, \tau)$-twisted Jackson type $q$-derivative.

### 5. Construction of $(q, \sigma, \tau)$-Differential Graded Algebra by Means of $(\sigma, \tau)$-Pre-Cosimplicial Algebra

In this section, we introduce the notion of $(\sigma, \tau)$-pre-cosimplicial algebra and show that a $(q, \sigma, \tau)$-differential algebra can be constructed with the help of a $(\sigma, \tau)$-pre-cosimplicial algebra.

First, we recall the notion of a pre-cosimplicial vector space [20]. A pre-cosimplicial vector space $(\mathfrak{A}, \mathfrak{f}_i)$ is a positive graded vector space $\mathfrak{A} = \oplus_{n \geq 0} \mathfrak{A}^n$ together with coface homomorphisms (linear mappings of vector spaces) $\mathfrak{f}_i : \mathfrak{A}^n \to \mathfrak{A}^{n+1}$, where $i$ runs from zero to $n + 1$, such that:

$$\mathfrak{f}_j \circ \mathfrak{f}_i = \mathfrak{f}_i \circ \mathfrak{f}_{j-1}, \quad i < j. \tag{15}$$

Thus, every pair of vector spaces $\mathfrak{A}^n, \mathfrak{A}^{n+1}$ is equipped with the $n + 2$ coface homomorphisms $\mathfrak{f}_0, \mathfrak{f}_1, \ldots, \mathfrak{f}_{n+1}$, where $\mathfrak{f}_i : \mathfrak{A}^n \to \mathfrak{A}^{n+1}$. For example, in the case of vector spaces $\mathfrak{A}^1, \mathfrak{A}^2$, there are three coface homomorphisms $\mathfrak{f}_0, \mathfrak{f}_1, \mathfrak{f}_2 : \mathfrak{A}^1 \to \mathfrak{A}^2$, which satisfy:

$$\mathfrak{f}_1 \circ \mathfrak{f}_0 = \mathfrak{f}_0^2, \quad \mathfrak{f}_2 \circ \mathfrak{f}_0 = \mathfrak{f}_0 \circ \mathfrak{f}_1, \quad \mathfrak{f}_2 \circ \mathfrak{f}_1 = \mathfrak{f}_1^2.$$

The following definition generalizes the notion of a pre-cosimplicial algebra, which can be found in [18].

**Definition 3.** *Let $\sigma, \tau$ be two degree zero endomorphisms of a pre-cosimplicial vector space $(\mathfrak{A}, \mathfrak{f}_i)$ such that they commute with coface homomorphisms, i.e.,:*

$$\sigma \circ \mathfrak{f}_i = \mathfrak{f}_i \circ \sigma, \quad \tau \circ \mathfrak{f}_i = \mathfrak{f}_i \circ \tau.$$

*A pre-cosimplicial vector space $(\mathfrak{A}, \mathfrak{f}_i)$ is said to be a $(\sigma, \tau)$-pre-cosimplicial algebra if:*

*(1)   $\mathfrak{A} = \oplus_{n \geq 0} \mathfrak{A}^n$ is a graded algebra,*
*(2)   $\sigma, \tau$ are degree zero endomorphisms of a graded algebra $\mathfrak{A}$,*
*(3)   for any homogeneous elements $u, v \in \mathfrak{A}$ and any integer $i \in \{0, 1, \ldots, |u| + |v| + 1\}$, we have:*

$$\mathfrak{f}_i(uv) = \begin{cases} \mathfrak{f}_i(u)\, \tau(v), & \text{if } |u| \geq i, \\ \sigma(u)\, \mathfrak{f}_{i-|u|}(v), & \text{if } 0 \leq |u| < i, \end{cases} \qquad \mathfrak{f}_{|u|+1}(u)\tau(v) = \sigma(u)\, \mathfrak{f}_0(v). \tag{16}$$

In particular, if we take $\sigma = \tau = \mathrm{id}_{\mathfrak{A}}$, then the above definition reduces to the definition of a pre-cosimplicial algebra.

**Theorem 2.** *Let $(\mathfrak{A}, \mathfrak{f}_i)$ be a $(\sigma, \tau)$-pre-cosimplicial algebra. Define the degree one linear mapping $d : \mathfrak{A}^n \to \mathfrak{A}^{n+1}$ by:*

$$d = \sum_{k=0}^{n} q^k\, \mathfrak{f}_k - q^n\, \mathfrak{f}_{n+1}, \tag{17}$$

*where $q$ is a primitive $N^{th}$ root of unity. Then, a $(\sigma, \tau)$-pre-cosimplicial algebra $\mathfrak{A}$ endowed with the degree one linear mapping $d$ is the $(q, \sigma, \tau)$-differential graded algebra.*

**Proof.** According to Definition 3, degree zero endomorphisms $\sigma, \tau$ of a positive graded vector space $\mathfrak{A}$ commute with coface homomorphisms $\mathfrak{f}_i$, and this immediately implies that $\sigma, \tau$ commute with $d$. It is proven in [16] that for any pre-cosimplicial vector space $\mathfrak{A}$, the degree one linear mapping $d$, defined in (17), satisfies $d^N = 0$, i.e., according to the terminology adopted in [16], $d$ is $N$-differential. Since the right-hand side of the formula for $d$ does not depend on degree zero endomorphisms $\sigma, \tau$, the same result holds in the case of $(\sigma, \tau)$-pre-cosimplicial algebra. Hence we only need to prove that $d$ satisfies the $(q, \sigma, \tau)$-Leibniz rule.

Let $u \in \mathfrak{A}^k$ be a homogeneous element of degree $k$ and $v \in \mathfrak{A}$. Our aim is to prove:

$$d(uv) = d(u)\,\tau(v) + q^k\,\sigma(u)\,d(v). \tag{18}$$

Making use of Formulas (16) and (17), we can write the left-hand side of the $(q, \sigma, \tau)$-Leibniz rule as follows:

$$
\begin{aligned}
d(uv) &= \mathfrak{f}_0(uv) + q\,\mathfrak{f}_1(uv) + \ldots + q^k\,\mathfrak{f}_k(uv) + q^{k+1}\,\mathfrak{f}_{k+1}(uv) + \ldots + q^n\,\mathfrak{f}_n(uv) - q^n\,\mathfrak{f}_{n+1}(uv) \\
&= f_0(u)\,\tau(v) + q\,f_1(u)\,\tau(v) + \ldots + q^k\,f_k(u)\,\tau(v) + q^{k+1}\sigma(u)\mathfrak{f}_1(v) \\
&\qquad\qquad\qquad + \ldots + q^n\,\sigma(u)f_{n-k}(v) - q^n\,\sigma(u)f_{n+1-k}(v).
\end{aligned} \tag{19}
$$

The right-hand side of the same formula can be written in the form:

$$
\begin{aligned}
d(u)\,\tau(v) \;+\; q^k\,\sigma(u)\,d(v) &= \mathfrak{f}_0(u)\,\tau(v) + q\,\mathfrak{f}_1(u)\,\tau(v) + \ldots + q^k\,\mathfrak{f}_k(u)\,\tau(v) - \cancel{q^k\,\mathfrak{f}_{k+1}(u)\tau(v)} \\
&\quad + \cancel{q^k\,\sigma(u)\mathfrak{f}_0(v)} + q^{k+1}\sigma(u)\mathfrak{f}_1(v) + \ldots + q^{k+l}\sigma(u)\mathfrak{f}_{n-k}(v) - q^{k+l}\sigma(u)\mathfrak{f}_{n+1-k}(v),
\end{aligned} \tag{20}
$$

where the crossed out terms cancel each other because of the second relation in (16). Comparing (19) with (20), we see that their left-hand sides are equal, and this ends the proof. $\square$

Let $\mathcal{A}$ be an associative unital algebra, whose unit element will be denoted by **1**, and $\sigma, \tau$ be two endomorphisms of $\mathcal{A}$. The tensor algebra $\mathfrak{T}(\mathcal{A}) = \oplus_{n \geq 0} \mathfrak{T}^n(\mathcal{A})$ is the graded algebra, where a subspace of elements of degree $n$ is the tensor product $\otimes^{n+1}\mathcal{A}$, i.e., $\mathfrak{T}^n(\mathcal{A}) = \otimes^{n+1}\mathcal{A}$, and the algebra multiplication $(u, v) \to uv$, where $u = u_0 \otimes u_1 \otimes \ldots \otimes u_n$ and $v = v_0 \otimes v_1 \otimes \ldots \otimes v_m$ are homogeneous elements of degree $n$ and $m$, respectively, is defined by:

$$uv = u_0 \otimes u_1 \otimes \ldots \otimes u_{n-1} \otimes u_n v_0 \otimes v_1 \otimes \ldots \otimes v_m. \tag{21}$$

We extend endomorphisms $\sigma, \tau$ to the tensor algebra $\mathfrak{T}(\mathcal{A})$ by:

$$\sigma(u) = \sigma(u_0) \otimes \sigma(u_1) \otimes \ldots \otimes \sigma(u_n), \quad \tau(u) = \tau(u_0) \otimes \tau(u_1) \otimes \ldots \otimes \tau(u_n).$$

Obviously, $\sigma, \tau$ are degree zero endomorphisms of graded algebra $\mathfrak{T}(\mathcal{A})$.

**Theorem 3.** *For any $u = u_0 \otimes u_1 \otimes \ldots \otimes u_n \in \mathfrak{T}^n(\mathcal{A})$, define the linear mappings $\mathfrak{f}_k : \mathfrak{T}^n(\mathcal{A}) \to \mathfrak{T}^{n+1}(\mathcal{A})$, where $k \in \{0, 1, \ldots, n+1\}$, by:*

$$
\begin{aligned}
\mathfrak{f}_0(u) &= \mathbf{1} \otimes \tau(u_0) \otimes \tau(u_1) \otimes \ldots \otimes \tau(u_n), \\
\mathfrak{f}_k(u) &= \sigma(u_0) \otimes \sigma(u_1) \otimes \ldots \otimes \sigma(u_{k-1}) \otimes \mathbf{1} \otimes \tau(u_k) \otimes \ldots \otimes \tau(u_n), \quad k = 1, 2, \ldots, n-1, \\
\mathfrak{f}_{n+1}(u) &= \sigma(u_0) \otimes \sigma(u_1) \otimes \ldots \otimes \sigma(u_n) \otimes \mathbf{1}.
\end{aligned} \tag{22}
$$

*Then, $(\mathfrak{T}(\mathcal{A}), \mathfrak{f}_k)$ is the $(\sigma, \tau)$-pre-cosimplicial algebra, and $\mathfrak{f}_k$ are its coface homomorphisms. If we endow the $(\sigma, \tau)$-pre-cosimplicial algebra $(\mathfrak{T}(\mathcal{A}), \mathfrak{f}_k)$ with the N-differential $d$, defined in (17), then $(\mathfrak{T}(\mathcal{A}), \mathfrak{f}_k)$ becomes the $(q, \sigma, \tau)$-differential graded algebra.*

**Proof.** Let $u = u_0 \otimes u_1 \otimes \ldots \otimes u_n$, $v = v_0 \otimes v_1 \otimes \ldots \otimes v_n$ be two homogeneous elements of tensor algebra $\mathfrak{T}(\mathcal{A})$. Then, their product:

$$uv = u_0 \otimes u_1 \otimes \ldots \otimes u_{n-1} \otimes u_n v_0 \otimes v_1 \otimes \ldots \otimes v_m$$

is the element of the subspace $\mathfrak{T}^{n+m+1}(\mathcal{A})$, and consequently, we have $\mathfrak{f}_0, \mathfrak{f}_1, \ldots, \mathfrak{f}_{n+m+1}$ coface homomorphisms from the subspace $\mathfrak{T}^{n+m}(\mathcal{A})$ to the subspace $\mathfrak{T}^{n+m+1}(\mathcal{A})$. For the coface homomorphisms $\mathfrak{f}_0, \mathfrak{f}_1, \ldots, \mathfrak{f}_n$, we have to prove the formula:

$$\mathfrak{f}_k(uv) = \mathfrak{f}_k(u)\,\tau(v), \; k = 0, 1, 2, \ldots, n. \tag{23}$$

According to the definition of coface homomorphisms, we can write the left-hand side as follows:

$$\mathfrak{f}_k(uv) = \sigma(u_0) \otimes \ldots \otimes \sigma(u_{k-1}) \otimes \mathbf{1} \otimes \tau(u_k) \otimes \ldots \otimes \tau(u_n v_0) \otimes \ldots \otimes \tau(v_m).$$

The right-hand side can be written as follows:

$$\mathfrak{f}_k(u)\tau(v) = \sigma(u_0) \otimes \ldots \sigma(u_{k-1}) \otimes \mathbf{1} \otimes \tau(u_k) \otimes \ldots \otimes \tau(u_n)\tau(v_0) \otimes \ldots \otimes \tau(v_m).$$

Since $\tau$ is an endomorphism of algebra, we have $\tau(u_n v_0) = \tau(u_n)\tau(v_0)$, and Formula (23) is proven. For coface homomorphisms $\mathfrak{f}_{n+1}, \ldots, \mathfrak{f}_{n+m+1}$, we have to prove:

$$\mathfrak{f}_k(uv) = \sigma(u)\mathfrak{f}_{k-n}(v), \; k = n+1, \ldots, n+m+1. \tag{24}$$

The proof of this formula is similar to the proof of (23). Finally, we have to prove the second relation in (16), i.e.,

$$\mathfrak{f}_{n+1}(u)\tau(v) = \sigma(u)\mathfrak{f}_0(v).$$

The left-hand side of this relation can be written as:

$$\mathfrak{f}_{n+1}(u)\tau(v) = \sigma(u_0) \otimes \sigma(u_1) \otimes \ldots \otimes \sigma(u_n) \otimes \mathbf{1}\,\tau(v_0) \otimes \ldots \otimes \tau(v_m)),$$

and the right-hand side can be written as:

$$\sigma(u)\mathfrak{f}_0(v) = \sigma(u_0) \otimes \sigma(u_1) \otimes \ldots \otimes \sigma(u_n)\,\mathbf{1} \otimes \tau(v_0) \otimes \ldots \otimes \tau(v_m)),$$

and we see that they are equal. □

**Author Contributions:** Investigation, V.A., O.L. and A.M.; Writing-Original Draft Preparation, V.A.

**Funding:** The first two authors gratefully acknowledge that this work was financially supported by the institutional funding IUT20-57 of the Estonian Ministry of Education and Research. The first author also gratefully acknowledges the financial support of the Département de Mathématiques, Université de Haute-Alsace, within the framework of the visiting professor program.

**Acknowledgments:** We thank the reviewer of our paper for valuable suggestions, particularly for drawing our attention to the term Weyl pair, which is used for matrices generating the matrix representation of the generalized Clifford algebra with two generators. The first author also expresses his gratitude for the invitation to visit the Département de Mathématiques, Université de Haute-Alsace, for hospitality during his stay, and for a very creative atmosphere while preparing this paper.

**Conflicts of Interest:** The authors declare no conflict of interest.

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
