# Peer review of "(q, σ, τ)-Differential Graded Algebras"

_universe, doi:10.3390/universe4120138_

Round 1

Reviewer 1 Report

This is an interesting paper on a generalization of the notions of (s,t)-differential graded algebra and q-differential graded algebra. The two known notions are combined to give rise to a new notion of (q,s,t)-differential graded algebra. The notion of (q,s,t)-differential graded algebra is illustrated through two examples: one derived from a generalized Clifford algebra and the other from a pre-co-simplicial algebra.

A complete check of the proofs in the whole paper would require several weeks. However, I did not see any incoherence in the presented results and I can do the following remarks and suggestions.

(i) The formalism is developed under the hypothesis that q is a primitive Nth root of unity. This is certainly the best choice as far as mathematics (and more specifically the representation theory) are concerned. From a physical point of view, there are plenty of q-deformed physical models for which q is a positive real parameter lower than 1. From a mathematical point of view, I am wondering if (q,s,t)-differential graded algebras might be of interest for 0 < q < 1. Could the authors say a few words about this question?  

(ii) It seems logical that a (q,s,t)-differential graded algebra yields a (s,t)-differential graded algebra in the limit case where q goes to 1 (N going to infinity). This point should be addressed in the paper.

(iii) In the mathematical physics literature, the two Nth order matrices in Eq. (10) are refereed to as a Weyl pair (in the present days they are largely used in quantum information). I suggest that the authors denote the two Nth order matrices in Eq. (10) as a Weyl pair.

Finally, I detected two misprints. First, on the third line after the listing of the authors: "mathématigues" should be replaced by "mathématiques". Second, on line 64, a reference should replace "??".

Author Response

Our point-by-point response to reviewer's comments is in uploaded file.

Reviewer 2 Report

     In the paper the authors introduce the notion of (q,\sigma,\tau) differential graded algebra, where q is a root of unity. They extend the existing notions, in the literature, of (\sigma,\tau) differential graded algebra and q-differntial graded algebra (see the cited references). They present some examples of their construction. They further introduce the notion of (\sigma,\tau) pre-cosimplicial algebra. 

     The paper is clearly written. The results are sound and interesting. It deserves being published as is.

Author Response

We agree with the opinion of the reviewer 2.